# Advances in Oncolytic Viral Therapy in Melanoma: A Comprehensive Review

**DOI:** 10.3390/vaccines13070727

**Published:** 2025-07-03

**Authors:** Ayushi Garg, Rohit Rao, Felicia Tejawinata, Gazi Amena Noor Shamita, McKay S. Herpel, Akihiro Yoshida, Gordon Goolamier, Jessica Sidiropoulos, Iris Y. Sheng, Salim-Tamuz Abboud, Luke D. Rothermel, Nami Azar, Ankit Mangla

**Affiliations:** 1Wayne State University School of Medicine, Detroit, MI 48201, USA; ayushigarg@wayne.edu; 2Case Western Reserve University School of Medicine, Cleveland, OH 44106, USA; rohit.rao@uhhospitals.org (R.R.); fit4@case.edu (F.T.); gxs483@case.edu (G.A.N.S.); axy234@case.edu (A.Y.); luke.rothermel@uhhospitals.org (L.D.R.); 3University Hospitals Seidman Cancer Center, Cleveland, OH 44106, USA; mckay.herpel@uhhospitals.org (M.S.H.); gordon.goolamier@uhhospitals.org (G.G.); jessica.sidiropoulos@uhhospitals.org (J.S.); 4Case Comprehensive Cancer Center, Cleveland, OH 44106, USA; 5University Hospitals Cleveland Medical Center, Cleveland, OH 44106, USA; salim.abboud@uhhospitals.org (S.-T.A.); nami.azar@uhhospitals.org (N.A.)

**Keywords:** oncolytic viruses, melanoma, immunotherapy, history of oncolytic viruses, T-VEC, RP-1, vusolimogene oderparepvec, talimogene laherparepvec, adenovirus, vaccinia virus, Herpes Simplex Virus

## Abstract

Checkpoint inhibitor therapy revolutionized the treatment of patients with melanoma. However, in patients where melanoma exhibits resistance to checkpoint inhibitor therapy, the treatment options are limited. Oncolytic viruses are a unique form of immunotherapy that uses live viruses to infect and lyse tumor cells to release the elusive neoantigen picked up by the antigen-presenting cells, thus increasing the chances of an immune response against cancer. Coupled with checkpoint inhibitors, intratumoral injections of the oncolytic virus can help an enhanced immune response, especially in a tumor that displays resistance to checkpoint inhibitors. However, oncolytic viruses are not bereft of challenges and face several obstacles in the tumor microenvironment. From the historical use of wild viruses to the sophisticated use of genetically modified viruses in the current era, oncolytic virus therapy has evolved tremendously in the last two decades. Increasing the ability of the virus to select the malignant cells over the non-malignant ones, circumventing the antiviral immune response from the body, and enhancing the oncolytic properties of the viral platform by attaching various ligands are some of the several improvements made in the last three decades. In this manuscript, we trace the journey of the development of oncolytic virus therapy, especially in the context of melanoma. We review the clinical trials of talimogene laherparepvec in patients with melanoma. We also review the data available from the clinical trials of vusolimogene oderparepvec in patients with melanoma. Finally, we review the use of various oncolytic viruses and their challenges in clinical development. This manuscript aims to create a comprehensive literature review for clinicians to understand and implement oncolytic virus therapy in patients diagnosed with melanoma.

## 1. Introduction

Intratumoral injections have a long history of evolution. William Coley first reported the antitumoral activity of injecting heat-inactivated *Streptococcus pyogenes* and *Serratia marcescens* in patients with soft tissue and bone sarcoma [1]. With this treatment, known as Coley’s toxin, he reported tumor regression in several patients diagnosed with metastatic or unresectable sarcomas. Although initially ridiculed by the scientific community, the idea of intratumoral injections eliciting an immune response against cancer caught on almost a century later [2]. Amongst intratumoral therapies, oncolytic viruses (OVs) and cytokines are the frontrunners in the fight against cancer [3]. An OV selectively infects cancer cells, leading to cell lysis and releasing tumor-derived antigens. Antigen-presenting cells (APCs), like dendritic cells, recognize and process these antigens and present them to the T cells to mount an immune response [3,4]. The cytokines aid the priming of T cells towards these antigens and help them mobilize into the tumor microenvironment (TME). (Figure 1). Checkpoint inhibitors, either administered systemically or built into the construct of the intratumoral agent, add to the potency of these T cells, leading to the lysis of the tumor colony [4].

OV therapy has evolved over the last several decades. From the initial attempts at harnessing the naturally occurring viruses, like the West Nile virus (WNV), yellow fever virus, etc., to genetically engineered viruses that are not only adept at countering the hostile TME but also selectively infect the tumor cells, OV therapy has come a long way [4,5]. Direct intratumoral injection of an OV is better at generating an antitumoral immune response compared to systemic immune checkpoint inhibitors (ICIs) [6]. Tumors with a higher ‘tumor-infiltrating lymphocyte’ (TIL) concentration have a higher PD-L1 expression and a better response to ICIs [7]. However, the mere presence of TILs is insufficient to generate an immune response due to several suppressive factors within the TME. The complexities of TME are described comprehensively by Junttila et al [8]. The success of intratumoral OVs depends on the ability to infect the tumor cells, spare the normal cells, evade the antiviral immune response, and incite a systemic antitumor immune response (abscopal effect) to achieve a durable response [6]. In this review, we review the OVs, either approved or in development, for treating patients diagnosed with melanoma.

## 2. A Brief History of Oncolytic Viral Therapy

Viruses are super-predators with a small size, easy adaptability to their environment, and an endless ability to mutate and evolve. To view them as a partner in the fight against cancer was considered preposterous until early reports of two patients with Hodgkin’s disease going into brief remission after contracting hepatitis emerged in 1949. Viruses were first noticed in 1892 [9], but several reports since the mid-19th century reported improvement in cancer with natural infections (which were not known to be viral infections back then) [9]. The most popular case is of a 42-year-old woman with presumed myelogenous leukemia who went into remission with the presumed influenza infection in 1897 [10]. Since then, multiple case studies have shown a common theme regarding the effects of viral infections on tumors in humans [11,12,13,14]. Given the right clinical conditions, systemic viral infections could cause tumor regression, albeit for a brief period only, and the responses were usually noted when patients were young and immunocompromised.

The development of OVs took a decisive turn after the ex vivo culture of human cells became possible in 1948. Then, investigators could test the in vivo activity of antitumor antibodies and the activity of OV platforms. The first success was reported by Alice E. Moore in 1949 when she demonstrated successful regression of ‘mouse sarcoma 180’ with Russian encephalitis virus (REV) [15]. She replicated her findings in five different tumors but also noted that REV caused fatal encephalitis in all mice. In subsequent years, many viruses were investigated as OVs with mixed success [16]. The first trials with viruses as a treatment for cancer were performed by inoculating patients with cancer with fluids or tissue obtained from patients diagnosed with infections. The extract (serum or tissue) was administered intravenously (IV), intramuscularly (IM), or intratumorally. Several viruses, including hepatitis B, Egypt 101, adenovirus, mumps, vaccinia, and dengue virus, were used in various trials [14,17,18,19,20]. Although these trials were rather crude in nature and lacked adequate control, they did inform the scientific community of the challenges of oncolytic viral therapy. Where systemic infection was a known risk of these treatments (especially with unpurified extracts), the presence of pre-existing neutralizing antibodies in the serum did not allow for prolonged exposure to the OV, which led to short durations of treatment response, if one was achieved at all.

The potential for virulence and rapid development of immunity to human viruses led to the idea of exploring animal viruses as OVs. After an extensive search for a suitable candidate, 6 out of 24 potential candidate viruses were selected for trials [21]. Equine rhinopneumonitis and infectious bovine rhinotracheitis were two herpesviruses (HSVs) that were non-pathogenic to humans and possessed oncolytic properties [22]. Arenaviruses had robust activity in rodent models but performed dismally in clinical trials [22,23]. Avian viruses showed promising activity in rodents but demonstrated strong neurotropism, leading to the death of mice [24]. Almost all nonhuman viruses showed strong tropism to non-malignant tissue, leading to the concern of systemic infections, particularly encephalitis. In addition, some of these viruses evolved and gained virulent properties in previously non-susceptible hosts. One of the prominent examples is that of the feline panleukopenia virus, which evolved and became transmissible in dogs to become canine parvovirus, leading to a pandemic in wild and domestic dogs in 1978–1979 [25]. However, a few animal viruses showed promise as OV platforms. Newcastle disease virus (NDV) is an avian virus that showed significant oncolytic activity in rodents when inoculated intraperitoneally without causing a systemic infection [26]. Clinical trials with NDV oncolysate showed durable remission in patients with melanoma in both metastatic/unresectable and postsurgical settings [26,27,28,29,30]. Likewise, vesicular stomatitis virus (VSV), a bovine virus affecting domestic cattle, has no pathogenicity in humans due to centuries of exposure resulting in immunity [31]. However, VSV still retains oncolytic activity and is being actively explored in cancer therapy [32,33].

Considering the virulence of naturally derived viruses, attempts were made to attenuate and engineer viruses to render them non-pathogenic. Early in the 1920s, vaccines against smallpox and rabies could induce remission in rodent tumors, but clinical trials were unsuccessful [34,35]. By the early 1950s, it was observed that passing an OV several times through the same tumor would cause its oncolytic properties to increase manifold [35,36,37]. Although the progeny would lyse the cancer cells more efficiently, this process could not reduce the tropism to non-malignant cells. Experiments with viral interference in an attempt to reduce the tropism of an OV did not meet with success either. For instance, attempts to inoculate NDV intracerebrally to interfere with the tropism of other OVs rendered the OV ineffective, the most popular being the Egypt 101 strain of WNV [38]. The discovery of the reverse transcriptase enzyme in 1970 and advances in recombinant technology by 1990 opened the floodgates for the genetic modification of viruses. Marutza et al. reported the first genetically modified HSV, lacking thymidine kinase, which rendered it incapable of infecting quiescent cells (essentially the non-malignant tissue) [39]. Since then, genetically modified OVs have been a subject of interest in treating cancers. Two OV platforms, namely H101 (a recombinant oncolytic adenovirus, commercially known as Oncorine^®^) in China (for head and neck cancers) and talimogene laherparepvec (T-VEC) (a recombinant oncolytic HSV-1, commercially known as OncoVEXGM-CSF^®^) in the U.S. (for melanoma), have secured approvals from the respective Food and Drug Administration (FDA) agencies. Multiple other OV platforms are currently in clinical trials, with vusolimogene oderparepvec (VuSO) being the frontrunner in recent times [40].

## 3. Mechanism of Oncolytic Virus Therapy

The quintessential feature of an OV is its ability to infect tumor cells while sparing healthy cells. OVs can acquire this capability either naturally or via genetic manipulation. They have a dual mechanism of action by which they cause tumor cell lysis directly and activate a systemic tumor response [4,6]. The selection of an OV is not only based on the pathogenicity of the virus and specificity for the tumor cell but also based on its ability to evade an antiviral immune response, produce therapeutic genes, and achieve sufficient titers.

### 3.1. Selective Infection and Replication in Tumor Cells

OVs are engineered to increase the specificity of the virus to cancer cells (called tumor tropism) and evade antiviral responses. The capsid of an OV can be modified to attach to specific surface receptors on cancer cells preferentially. For example, an adenovirus utilizes the expression of Coxsackie and Adenovirus Receptor (CAR), integrins, and cluster of differentiation (CD) 46 [41]. Likewise, HSV-1 targets receptors like HVEM (Herpes Virus Entry Mediator) and nectin-1, and measles virus targets CD46 [42]. OVs also take advantage of the inherently dysfunctional antiviral responses in the tumor cells [43,44]. The selective deletion of virulent genes and insertion of promoter genes help OVs to replicate uninhibited in tumor cells [45]. The abundance of nucleotides in tumor cells and aberrant activation of oncogenic pathways help the attenuated viruses to replicate freely [4]. Several viruses, like HSV-1 (manipulation of ICP34.5), vaccinia virus (manipulation of thymidine kinase), and adenoviruses (deletion of E1A in RB gene mutated cells), function in this way [46,47,48]. Lastly, placing the viral genes under tissue-specific promoters helps an OV to replicate efficiently in tumor cells. Trials exploring this modality are underway with adenoviruses and HSV-1 [49,50].

### 3.2. Stimulating the Immune System to Cause Cell Death

Dendritic cells (DCs) are an integral part of the innate immune system and are critical in the antitumor response as APCs. OVs cause ‘immunogenic cell death’ (ICD), resulting in the release of the damage-associated molecular pattern (DAMP), which attracts DCs [51,52]. Compared to apoptotic cell death, which induces immunotolerance, the ICD induced by an OV activates the immune system [53]. In addition, a complex interplay of viral elements with Toll-like receptors (TLRs) and other innate receptors leads to the maturation of DCs, thus promoting the release of cytokines like interleukin-12 (IL-12), IL-6, IL-1β, and TNF, which help in the recruitment and priming of T cells [54,55]. The role of interferons (IFNs) in antitumor immunity is not clearly understood. However, IFN-1 increases antigen presentation and converts an innate immune response to an adaptive immune response [56]. The immune system is a double-edged sword in leveraging OVs for cancer-directed therapy. OVs evade a robust T-cell response by preventing their antigen from being processed. They are engineered to delete the infected cell protein (ICP) 47 gene, which is critical to trafficking the viral antigen into the endoplasmic reticulum, thus hindering antigen presentation [57,58]. On the other hand, viral antigens, being a potent immunogen, generate robust T-cell activation signals, recruiting T cells to the TME. In the milieu of the lysed tumor cells, soluble tumor-associated antigens are present, which can promote presentation of the tumor antigen to T cells, thus contributing to the antitumor activity [4].

### 3.3. Delivering Therapeutic Genes and Altering Tumor Neovasculature

OVs deliver genes that can either be cytotoxic or immunomodulatory. One prime example is the HSV-1 derivative T-VEC carrying the gene that produces human GM-CSF. GM-CSF regulates the function of DCs and aids in their maturation, thus aiding in the innate immune response [59]. OVs can also be modified to express tumor-associated antigens [60], T-cell co-stimulatory molecules [61], or OV-driven expression of T-cell checkpoint molecules, which can aid and abet in generating an adaptive immune response to the tumor antigen [4,6]. OVs can be engineered to alter the tumor neovasculature, thus inducing an inflammatory environment that elicits an antitumor response. VSV is a prime example of an OV that blocks the tumor neovasculature, leading to inflammation within the tumor microenvironment which leads to recruitment of neutrophils and formation of microthrombi [62].

## 4. Talimogene Laherparepvec: The First Oncolytic Viral Therapy

T-VEC is a genetically modified HSV-1 that selectively infects and lyses tumor cells while stimulating an immune response [63,64,65]. The deletion of the ICP34.5 and ICP47 genes enhances OV replication in cancer cells and antigen presentation, respectively [48,64,65,66]. T-VEC expresses GM-CSF, recruiting DCs to activate T cells to trigger local tumor destruction and systemic antitumor immunity in patients with unresectable melanoma [67].

### 4.1. T-VEC as Monotherapy

T-VEC was approved by the United States FDA (USFDA) in October 2015 as a monotherapy for patients who experienced recurrence with cutaneous lesions after surgery. T-VEC is not approved as intratumoral therapy for visceral lesions [63]. However, the approval for T-VEC came at a time when dual checkpoint inhibition with ipilimumab 3 mg/kg and nivolumab 1 mg/kg was also receiving approval (also approved in October 2015) [68]. Since then, the role of T-VEC has evolved, and several trials have explored the combination of T-VEC with ICIs [69,70].

In the phase I trial of T-VEC, 26 of 30 patients were evaluable [71]. There were no clinical responses, but three patients showed stable disease (SD). Of 26 patient post-treatment biopsies, 19 had residual tumors; 14 of these exhibited tumor necrosis, which was extensive in some cases. The overall clinical responses noted were as follows: six patients had tumor flattening, three had stable disease, and four patients showed inflammation in both injected and uninjected tumors [71]. The phase II trial reported a 26% overall response rate (ORR) and tumor regression in both injected and distant metastasis, including visceral metastasis [72]. The phase III OPTiM trial (N = 436) randomized patients in a 2:1 ratio to intralesional T-VEC (n = 295) versus subcutaneous GM-CSF (n = 141) only. Durable response rate (DRR), overall survival (OS), objective response rate (ORR), and duration of response (DoR) were recorded, amongst other endpoints. The T-VEC arm showed a trend towards better OS (23.3 vs. 18.9 months, unstratified hazard ratio (HR), 0.79; 95% confidence interval (CI), 0.62–1.00; P = 0.0494 [descriptive]) and a better DRR (19.0% vs. 1.4%, unadjusted odds ratio (OR), 16.6; 95% CI, 4.0–69.2; P < 0.0001) [73]. Fifty patients (16.9%) achieved a complete response (CR) in the T-VEC arm compared to one patient (0.7%) in the GM-CSF arm. Notably, 88.5% of patients who achieved a CR were alive at the 5-year landmark analysis, demonstrating the durability of the response [73]. Based on these results, T-VEC became the first OV therapy approved by the USFDA in 2015 [63].

A meta-analysis of eight studies of T-VEC, including 642 patients with stage IIIB-IVM1a melanoma, reported a complete response rate (CRR) of 30% and an ORR of 44% with T-VEC in patients, particularly those with injectable cutaneous lesions and with nodal metastases [74]. A study from the Netherlands Cancer Institute (N = 26) in patients with unresectable stage IIIB/C and stage IV(M1a only) melanoma (median follow-up: 12.5 months) showed an ORR of 88.5% and a disease control rate (DCR) of 92.3%. Complete clinical response was recorded in 61.5% of all patients. Treatment-related adverse events (TRAEs) occurred in all patients (100%). However, most patients experienced only mild adverse events. Grade 3–4 TRAEs occurred in only one patient who developed grade-3 pancolitis after three cycles of therapy, which responded to high-dose steroids, thus allowing treatment resumption safely without further serious adverse events [75]. The COSMUS-1 study (N = 76) reported a 19.7% CR in patients treated with T-VEC with no remaining injectable lesions. The most common side effects were flu-like symptoms (10.5%) and ulceration (5.3%) [76]. A COSMUS-2 (N = 83) trial reported that T-VEC may be used concurrently with or after anti-PD-1 therapy. Treatment discontinuation was noted in 25% of patients due to lesion clearance and 37% due to disease progression [77]. Retrospective studies have consistently shown a benefit of T-VEC in patients with stage IIIB/IVM1a melanoma [76,77,78].

### 4.2. T-VEC with Pembrolizumab in Treatment-Naïve Advanced Melanoma

The MASTERKEY-265 phase Ib trial was an open-label trial combining T-VEC with pembrolizumab in patients with unresectable stage IIIB-IVC melanoma. Twenty-one patients were enrolled and followed for a median of 58.6 months (interquartile range (IQR): 1.4–61.6 months). At the time of data cutoff in March 2020, nine patients who achieved a CR and three of four patients who achieved a PR remained in response, demonstrating the durability of the response. The OS and progression-free survival (PFS) were better for the responders than for those who did not respond (P = 0.0056) [79]. The promising results of the phase Ib trial led to a randomized, double-blind, placebo-controlled, global phase III trial comparing the combination of T-VEC with pembrolizumab with pembrolizumab monotherapy in patients with ICI-naïve, unresectable stage IIIB-IVC melanoma [80]. The primary endpoint was PFS. In the primary analysis, the median PFS was 14.3 months in the T-VEC–pembrolizumab arm compared to 8.5 months in the pembrolizumab-only arm (overall stratified HR, 0.86; 95% CI, 0.71 to 1.04; P = 0.13) [80]. While the PFS favored the T-VEC–pembrolizumab arm in three predefined subgroups: patients enrolled in the U.S. (HR 0.59; 95% CI, 0.37 to 0.92), those with baseline LDH ≤ upper limit of normal (HR 0.76, 95% CI, 0.59 to 0.99), and those with baseline sum of the longest diameters of target lesions ≤ median (HR 0.70, 95% CI, 0.51 to 0.96), no OS benefit was observed (HR 0.96; 95% CI, 0.76 to 1.22; P = 0.74). The final analysis, with a median follow-up of 25.58 months, confirmed no significant PFS or OS improvements with the combination of T-VEC and pembrolizumab [80]. TRAEs (most commonly, pyrexia and fatigue) were comparable in T-VEC–pembrolizumab and pembrolizumab-only arms (98.3% versus 96.2%). Grade ≥ 3 TRAEs occurred in 20.3% of patients in the combination arm and 15.7% in the control arm. Fatal TRAEs occurred in 1.2% of the intervention arm and 0.3% in the control arm, with 7.8% and 8.2% of patients dying from progressive disease in the respective arms [80]. The median PFS of 8.5 months in the pembrolizumab control arm was similar to the 8.4 months observed in the KEYNOTE-006 trial [81]. Although the phase III trial did not meet its primary endpoint, the PFS in the T-VEC–pembrolizumab combination arm was numerically superior (14.3 vs. 8.5 months). Coupled with a low rate of grade 3 or higher TRAEs and a numerically superior PFS, the combination of T-VEC and pembrolizumab is still considered a worthy option in select patients.

### 4.3. T-VEC with Ipilimumab in Treatment-Naïve Advanced Melanoma

In a phase Ib trial of T-VEC with ipilimumab in patients with unresectable stage IIIB-IVM1c melanoma, 19 patients were enrolled and included in the safety analysis. No dose-limiting toxicities were observed, and grade 3 or higher toxicity was noted in 26.3% of patients only. The ORR determined by immune-related response criteria was 50% (95% CI, 26.0 to 74.0), and the DRR was 44%. Four patients had CR, five had PR, and four had SD. Of all the responders, only one patient experienced the recurrence of melanoma within six months. Of the 35 injected, measurable lesions, 11 regressed completely, and an additional 15 lesions regressed by more than 50%. Of the 23 uninjected, measurable lesions, 9 regressed completely, and an additional 3 lesions had more than 50% regression. Seven of thirteen non-visceral and two of ten visceral uninjected measurable lesions regressed completely [82]. These results led to an open-label, multicenter, randomized phase II trial randomizing 198 patients to receive either the combination of T-VEC and ipilimumab or ipilimumab monotherapy. The primary endpoint was ORR, which was significantly better with the combination of T-VEC–ipilimumab compared to ipilimumab monotherapy (39% vs. 18%, 95% CI, 1.5 to 5.5; P = 0.002). Responses were noted in the uninjected visceral lesions in 52% of patients in the combination arm and 23% in the ipilimumab arm. The incidence of grade 3–4 TRAEs was comparable, with 45% of patients experiencing grade 3/4 AEs in the combination arm and 35% of patients in the ipilimumab monotherapy arm [83]. A five-year analysis showed that the combination of T-VEC and ipilimumab performed better than ipilimumab in terms of ORR (35.7% vs. 16.0%; OR 2.9; 95% CI 1.5 to 5.7; P = 0.003), DRR (33.7% vs. 13%, unadjusted OR 3.4; 95% CI 1.7 to 7.0; descriptive P = 0.001), and median PFS (13.5 vs. 6.4 months, HR 0.78; 95% CI 0.55 to 1.09; descriptive P = 0.14). However, the difference in median OS between the two arms was not statistically significant (54.7 vs. 48.4 months, HR 6.3, 95% CI: −8.1 to 20.7) [70]. The combination of T-VEC with ipilimumab is endorsed by the NCCN, but did not receive formal approval from the USFDA.

### 4.4. Neoadjuvant T-VEC with Nivolumab

The phase II NIVEC trial (N = 24) assessed the safety and efficacy of neoadjuvant nivolumab and T-VEC in patients with resectable stage IIIB-IVA melanoma (median age: 67 years) [84]. Of the patients, 58% had stage IIIB, 33% had IIIC, and 9% had stage IIID melanoma. Forty-six percent of patients had BRAF V600^E/K^-positive melanoma. Patients received four intralesional T-VEC doses and three nivolumab doses (240 mg every two weeks) concurrently before surgery [84]. The pathologic ORR was 74%, with a major pathological response noted in 65% of patients [84]. The 1-year event-free survival (EFS) was 75% (95% CI: 0.55–1). Grade 2 and 3 TRAEs occurred in 29% and 8% of patients, respectively, with no grade 4–5 toxicities [84].

These results were supported by a phase II randomized control trial (N = 150), where patients with resectable stage IIIb-IVM1a melanoma were randomized to receive either neoadjuvant T-VEC (six doses) followed by surgery or surgery alone [85]. At a median follow-up of 63.3 months, significant improvement was observed in the arm receiving neoadjuvant T-VEC compared to the surgery-alone arm. The 5-year recurrence-free survival (RFS) (22.3% vs. 15.2%; HR, 0.76; 80% CI, 0.60–0.97), 5-year EFS (43.7% vs. 27.4%; HR, 0.57; 80% CI, 0.43–0.76) and OS (77.3% vs. 62.7%; HR, 0.54; 80% CI, 0.36–0.81) favored the neoadjuvant TVEC arm. No new toxicity signals were identified [86]. The authors concluded that neoadjuvant administration of T-VEC is safe and effective and can lead to durable responses.

### 4.5. T-VEC in Combination with Pembrolizumab in Advanced Melanoma Refractory to Anti-PD1 Therapy

The MASTERKEY-115 trial was a phase II, open-label, multicenter trial that evaluated the combination of T-VEC and pembrolizumab in patients with advanced melanoma in whom the melanoma was refractory to anti-programmed cell death protein-1 (PD-1) inhibitors [69]. Seventy-two patients were randomized into four cohorts—Cohort 1 included patients with primary resistance to anti-PD-1 therapy, Cohort 2 included patients who had an initial clinical benefit from anti-PD-1 therapy but developed progressive diseases within 12 weeks of their last dose, Cohort 3 and Cohort 4 included patients who had resectable disease and received adjuvant anti-PD1 monotherapy and progressed within 6 months (Cohort 3) or after 6 months (Cohort 4) of completing the adjuvant anti-PD1 monotherapy. The combination treatment showed significantly better ORR in Cohort 3 (40.0%) and 4 (46.7%) compared to Cohort 1 (0%) and 2 (6.7%). The immune-related DCR (sum of CR + PR + SD) was 50.0% in Cohort 1, 40% in Cohort 2, 73.3% in Cohort 3, and 86.7% in Cohort 4, and PFS was highest in Cohort 3 and 4, with the median PFS not being reached. TRAEs occurred in 76.1% of patients, with 12.7% experiencing grade 3 or higher TRAEs. Additionally, 10 patients (14.1%) experienced treatment-emergent adverse events (defined as AEs occurring within 30 days of the last administration of the study treatment), and one patient had a fatal TRAE [69]. These results underscore the effectiveness of T-VEC combined with pembrolizumab in patients with melanoma in whom the tumor is resistant to anti-PD1 monotherapy.

## 5. Vusolimogene Oderparepvec: The Novel Drug

VuSo is a genetically modified OV derived from a new strain of HSV-1 (RH018), which has a superior oncolytic activity determined by in vitro studies [63,87]. Also known as RP-1, VuSo is enhanced with fusogenic glycoprotein gibbon ape leukemia virus surface glycoprotein with R-sequence deleted (GALV-GP-R-) and a codon-optimized sequence of GM-CSF [88]. The insertion of GALV-GP-R- increases the potency of RP-1 by increasing the expression of a fusogenic glycogenic protein on the surface of infected cells. This induces an increased fusion of infected cells, resulting in the formation of a syncytium, which is highly immunostimulatory [89]. In combination with GM-CSF, the GALV-GP-R- sequence enhances oncolysis and the systemic immune response [87]. RP-1 was investigated as a monotherapy and in combination with nivolumab in patients who did not respond to anti-PD1 monotherapy. The phase I/II IGNYTE study (NCT03767348) evaluated RP-1 in 156 patients with advanced melanoma who had at least one measurable and injectable tumor (≥1 cm) and had experienced disease progression after receiving anti-PD1 monotherapy, with or without anti-CTLA4 therapy, for a minimum of eight weeks [90]. RP-1 was initially administered intratumorally at a dose of 1 × 10^6^ plaque-forming units (PFU)/mL, followed by 1 × 10^7^ PFU/mL every two weeks for up to eight cycles, concurrently with nivolumab (240 mg IV every 2 weeks in cycles 2–8) [90]. After completing the initial treatment phase, patients continued nivolumab monotherapy (240 mg every two weeks or 480 mg every four weeks) for up to two years. The patients who had progressive or stable disease on restaging scans were allowed to receive additional intratumoral injections of RP-1 [90].

Within the cohort of treated patients, 46.2% had received prior anti-PD1 and anti-CTLA4 combination therapy, while 51.3% of patients had advanced stage IVM1b-d disease [90]. The ORR was 31.4%, with 12.2% of patients achieving a CR [90]. Responses were observed in treatment-naïve and heavily pretreated patients. An abscopal effect was noted in the non-injected lesions and bulky or visceral tumors as well [90]. The ORR was 34.1% in patients in whom the melanoma displayed anti-PD1 resistance and 26.4% in those previously treated with ipilimumab plus nivolumab [90]. The median DoR exceeded 24 months, with no relapse/progression seen in responders in the first 6 months, and 78% of patients were still in response at the last data cutoff [90]. Adverse events related to RP1 and nivolumab were mainly mild to moderate (grade 1–2), though one grade 5 case of immune-mediated myocarditis was reported [90]. Overall, RP-1, in combination with nivolumab, offered a durable and meaningful antitumor activity in patients with anti-PD1-resistant melanoma. In addition, durable responses were noted in both target (injected lesion) and non-target (or non-injected lesions) lesions, including visceral lesions, a significant development for intratumoral OV therapy [90].

A randomized, controlled, multicenter, phase III IGNYTE-3 trial (NCT06264180) is currently underway to evaluate the efficacy of RP1 in combination with nivolumab in patients with relapsed or refractory metastatic or locally advanced unresectable melanoma who have previously received the combination of anti-CTLA4 and anti-PD-1, and BRAF-directed treatment (in patients with BRAF mutated melanoma), or who are ineligible for anti-CTLA-4 treatment [91]. The study randomizes patients between those receiving intratumoral RP-1 along with nivolumab IV every two weeks for up to eight cycles, followed by nivolumab maintenance, and those receiving standard therapy selected by the physician [91]. The patients in the control arm can receive any approved anti-PD1 monotherapy (nivolumab or pembrolizumab), nivolumab combined with relatlimab, or chemotherapy (dacarbazine, temozolomide, or paclitaxel) [91]. OS is the primary endpoint, and PFS and ORR are the secondary endpoints [91]. Due to the positive results from the IGNYTE phase I/II trial, the combination of RP-1 and nivolumab received a breakthrough designation in late 2024 and a priority review from the USFDA.

RP-2 is an enhanced-potency, HSV-1-based OV with the addition of GALV-GP-R- (enhances cell fusion and tumor lysis), an anti-CTLA4 antibody-like molecule (improves T-cell priming and activation), and a codon-optimized sequence of GM-CSF (for recruiting and activating DCs) [63,92]. Additionally, the deletion of the ICP47 gene boosts antigen presentation, resulting in enhanced infiltration and recognition of tumor cells by cytotoxic T lymphocytes leading to effective tumor cell destruction [92,93,94]. In the phase I trial, objective responses with RP-2 monotherapy were observed in three out of nine patients—one CR (mucoepidermoid carcinoma) and two PRs (one in metastatic esophageal carcinoma and one in metastatic uveal melanoma) [93]. In the same trial, RP-2 combined with nivolumab had an ORR of 35% (four of nine patients with cutaneous melanoma, two of eight patients with uveal melanoma, and one of three patients with squamous cell carcinoma of the head and neck) [93]. RP-2 is currently under investigation as a monotherapy and in combination with nivolumab in various tumor types. A separate presentation at ASCO 2024 reported the results of the phase I trial of RP-2 in patients with uveal melanoma [94]. Seventeen patients with uveal melanoma, most of them heavily pretreated (including dual checkpoint inhibitor therapy), were included in this study. Three patients received RP-2 monotherapy, and fourteen patients received RP-2 with nivolumab. One response was noted in the monotherapy cohort, and four were noted in the combination cohort (all PR). Disease control (CR + PR + SD) was noted in 10 of 17 patients, with a median DoR of 11.5 (2.8–21.2) months. Two patients receiving the combination developed hypotension (grade 3), and no grade 4–5 adverse events were reported [94]. RP-2 is currently under investigation in a phase II/III trial in patients with metastatic uveal melanoma who have never received immune checkpoint inhibitor therapy (NCT06581406).

RP-3 is a further advancement of RP-1 and RP-2. In addition to expressing GALV-GP-R- and an anti-CTLA4 antibody, RP-3 also expresses 4-1BB- and CD40-activating ligands. RP-3 does not express GM-CSF [63,95,96]. The CD40 and 4-1BB ligands act as co-stimulatory molecules in activating the immune system. RP-3 deploys strategies to encompass innate and adaptive immune systems to tackle cancer. It is currently not explored in melanoma, likely because studies on RP-1 and RP-2 are in an advanced stage.

## 6. Oncolytic Viruses in Clinical Development for Patients with Melanoma

Viruses are versatile beings that are near-perfect predators. They can exist as extracellular virion particles, resistant to physical stress but vulnerable to humoral immunity, or as intracellular genomes with limited gene expression, evading host immunity [97]. Advances in gene editing coupled with the versatility of the viral platforms have led to the development of several viral platforms as oncolytic therapies. Many of these platforms have been investigated in patients with melanoma and other cancers. OVs are either DNA (deoxyribonucleic acid) or RNA (ribonucleic acid) viruses. Single-stranded RNA (ssRNA) and double-stranded DNA (dsDNA) viruses are the most commonly used viruses in developing OV platforms. Viruses can be attenuated naturally or altered genetically depending on their structures. DNA viruses hold an advantage over RNA viruses in terms of genetic stability and efficient replication, which makes them attractive for gene manipulation. On the contrary, due to their small size and genetic instability, RNA viruses are more immunogenic and demonstrate good adaptability for IV injections [6]. Table 1 lists the status of the current clinical trials exploring various OVs in patients with melanoma.

### 6.1. DNA Viruses

HSV, adenovirus, and vaccinia virus are some dsDNA viruses used to develop OV platforms. DNA-based vectors are ideal due to their large and stable genomic structures, which allow for easy manipulation.

#### 6.1.1. Herpesvirus

HSV is an enveloped virus with a 150 Kb dsDNA genome and a complex structure [105]. The envelope of HSV, aided by the glycoproteins gB, gD, gH, and gL, fuses with the lipid bilayer of the cell membrane and releases the genomic material into the cytosol [106]. The genomic material is transported to the nucleus, where transcription initiates, which is strictly regulated by the genome of HSV. This leads to the creation of viral proteins [107,108]. HSV is a cytolytic virus that can infect multiple types of cancer cells. It releases the progeny HSV in the TME, which infects other cancer cells [109]. HSV is also adept at evading host immunity despite the high prevalence of neutralizing antibodies. HSV-1 is the most commonly used strain for OV therapy, with T-VEC being the only approved HSV-1-based OV therapy platform for patients with melanoma. Other strains of HSV-1 (like G207, G47Δ) and HSV-2 (like OH2) are under investigation [110,111,112].

#### 6.1.2. Adenovirus

An adenovirus is a naked virus with a 26–45 kb dsDNA genome wrapped in an icosahedral capsid [113]. Over 57 different serotypes of adenoviruses are known. However, Ad2 and Ad5 are the most commonly used serotypes for the OV platform [114]. An adenovirus enters the host cell through receptor-mediated endocytosis and disassembles in the cytoplasm, releasing the capsid, which is then transported to the nuclear envelope via microtubules for internalization into the host nucleus. The E1A and E1B genes play a critical role in initiating transcription and preventing post-infection cell death, thus prolonging the cycle of viral replication [115,116,117]. An adenovirus offers a versatile OV platform because of its large genome, which allows the manipulation of genes and the ability to generate high viral titers and potent lytic activity. However, it has a significant disadvantage in exhibiting a strong potential to infect normal cells [118]. Several adenoviruses are under investigation, including ICOVIR5, which is currently under investigation in patients with melanoma. H101 is approved by the Chinese FDA for patients with nasopharyngeal carcinoma [119,120,121].

#### 6.1.3. Vaccinia Virus

Vaccinia virus is a brick-shaped virus with a 190 Kb dsDNA genome [122]. It is a complex virus with an asymmetrical and complex intracellular mature virion with an active life cycle and high rates of replication. The post-infection transcription process occurs exclusively in the cytoplasm and depends on the thymidine kinase. Vaccinia virus secretes viral proteins to activate the EGFR-RAS pathway of the host cells to promote the synthesis of thymidine kinase, which sustains the replication of vaccinia virus in the infected host cell [123]. Thymidine kinase is usually overexpressed in malignant cells and rarely expressed in normal cells. JX-594 (Pexa-vac) is the most famous vaccinia virus, and it was tested in patients with melanoma, but the clinical development of this virus has branched off to other cancers since then [124].

#### 6.1.4. Parvovirus

The *Parvoviridae* family includes 134 species, out of which 1 species—*Rodent protoparvovirus 1* (RoPV1) (genus *Protoparvovirus*)—can replicate autonomously in cancer cells. H-1 parvovirus (H-1PV) is a kind of RoPV1 that has shown immense potential as an OV therapy [125]. H-1PV is a naked virus with a 5.1 Kb ssDNA genome. The natural host of H-1PV is the rat, and it does not infect humans naturally [125]. H-1PV depends on the host cell for its survival. Cancer cells have inherent deficiencies like uncontrolled proliferation, dysregulated signaling pathways, and impaired innate immunity, which make them an ideal breeding ground for H-1PV [126]. Clinical trials with H-1PV in patients with glioblastoma multiforme (GBM) and pancreatic ductal adenocarcinoma are underway [127]. H-1PV-induced cell lysis stimulated DC activation and maturation in melanoma cell lines co-cultured with DCs [128].

### 6.2. RNA Viruses

RNA viruses are usually ssRNA viruses, except for reovirus, which is a dsRNA virus. ssRNA viruses can be positive-sense RNA viruses (genomic information is directly translated into proteins) and negative-sense RNA viruses (genomic information is first converted to positive-sense RNA viruses and then translated into proteins). RNA viruses are small and can cross the blood–brain barrier, which makes them attractive for treating CNS tumors (poliovirus for GBM). Although the instability in the genomic structure makes the RNA viruses more immunogenic, it also makes them hard to manipulate and more pathogenic. Reovirus, Coxsackievirus, Seneca Valley virus, NDV, poliovirus, and measles virus are being investigated as OV platforms [6].

#### 6.2.1. Reovirus

Reovirus is a naked, naturally occurring RNA virus with a 123 Kb dsRNA. Reovirus exploits the presence of junctional adhesion molecule A, which is overexpressed in multiple cancers (like breast cancer, non-small-cell lung cancer, diffuse large B-cell lymphoma, and multiple myeloma) to enter the host cell [129,130,131,132,133]. Reovirus also takes advantage of the mutation in RAS signaling (prevalent in multiple cancers), which inactivates PKR—a critical inhibitor of the translation of viral transcripts. Upon viral entry into the host, transcription and translation happen exclusively in the cytoplasm [134,135]. Three serotypes of reovirus have been identified, and amongst them, the type 3 Dearing strain is used to manufacture Reolysin [136]. Clinical trials in patients with multiple cancer types, including melanoma, are ongoing.

#### 6.2.2. Poliovirus

Poliovirus is a naked, positive-sense, ssRNA virus with a genome size of 7.5 Kb. It shows strong tropism towards cells expressing CD155, which is overexpressed in (neuro)ectodermal tumors [137]. Poliovirus depends on the internal ribosome entry site (IRES) element to initiate translation. Exchanging the IRES with an IRES of the related human rhinovirus type 2 helps attenuate the infectivity to primate cells while maintaining the pathogenicity of the malignant cells [138]. Lerapolturev (also known as PVS-RIPO) has been investigated in patients with GBM, melanoma, and bladder cancer with mixed results [139,140,141]. One of the significant concerns with PVS-RIPO is the instability of the genomic structure, which can lead to mutation in the OV, reverting it into a neuropathogenic virus [142].

#### 6.2.3. Coxsackievirus

Coxsackievirus is a naked, positive-sense, ssRNA virus with a genome size of 28 Kb. Coxsackievirus binds to the intercellular adhesion molecule 1 (ICAM-1) and needs decay-accelerating factor (DAF) attachment to enter the host cell [143]. Both ICAM-1 and DAF are overexpressed in melanoma cells, which led to the conception of a clinical trial with CVA-21 [144]. So far, the trial has shown promising results.

#### 6.2.4. Seneca Valley Virus

Seneca Valley virus (SVV) is a naked, positive-sense, ssRNA virus with a genome size of 7 Kb. SVV is found in bovine animals and is not pathogenic in humans [145]. However, it does infect malignant cells of neuroendocrine origin. A phase I trial of NTX-010 in patients with NE tumors demonstrated safety, and trials in several other tumor types are underway [146].

#### 6.2.5. Negative-Sense ssRNA Viruses

MeV and NDV, belonging to the family Paramyxoviridae, and VSV, belonging to the family Rhabdoviridae, are negative-sense ssRNA viruses that have been explored as OV platforms.

MeV is an enveloped ssRNA virus with an icosahedral capsid and a genome of 16 Kb. It is a highly contagious virus in its natural form [147]. Hence, the attenuated vaccine Edmonston strain is used as an OV platform. The overexpression of CD46, the receptor for MeV, on malignant cells makes it an attractive OV platform [148]. MV-NIS and MV-CEA are two OV platforms being tested in gynecologic cancers, GBM, and lung cancers [148,149]. NDV is an avian virus with no infective potential in humans. NDV is an enveloped ssRNA virus with a helical capsid and a genome of 15 Kb. PV-701 and NDV-HUJ are currently in clinical trials exploring efficacy and toxicity in multiple cancer types (including melanoma) [150]. Recombinant MeV and NDV are unique in having large genomic materials but short RNAs, which can accommodate the introduction of large foreign genomes.

VSV is an enveloped ssRNA virus with a helical capsid and a genome of 11 Kb. VSV has some distinct advantages over other OV platforms. It is not dependent on the cell cycle or a specific receptor for replication. It produces a high viral yield and has a small genome that can be easily manipulated [31]. There is no natural immunity in humans. However, it does not cause serious infections in humans either. Although VSV is not dependent on receptors, selectivity for cancer cells is increased by using the defects of the antiviral interferon signaling pathway in cancer cells [151]. The VSV-hIFNβ vaccine is in phase I clinical trials right now [152].

## 7. Conclusions

OV therapies have evolved for nearly a century but have taken long strides in the last three decades since the advent of recombinant technology. Genetically engineered OVs are more potent in bringing about oncolysis and exhibit a significant amount of tumor tropism and negligible risk of systemic infections. T-VEC is the only approved OV in the U.S. for patients with melanoma, but it can only be injected in cutaneous lesions. T-VEC is approved for use in patients with cutaneous lesions that recur after surgery. In our practice, we use T-VEC in patients with anti-PD-1 refractory melanoma who present with in-transit metastasis or cutaneous metastasis only. A multidisciplinary discussion should be held regarding the use of concurrent anti-PD1 or ipilimumab monotherapy based on the trials conducted so far. We do not support the use of TVEC monotherapy in the neoadjuvant setting due to positive results from the SWOG-1801 (pembrolizumab monotherapy) and NADINA (neoadjuvant ipilimumab and nivolumab) trials [153,154]. Several other OVs, like RP-1, -2, and -3, show promising results. Continuous improvement in the construct of these platforms not only improves their ability to infect and replicate preferentially in tumor cells but also allows them to incite an innate and adaptive immunity against tumor cells. The trials investigating the combination of T-VEC with checkpoint inhibitors have not met their primary endpoints. However, the preliminary data from the clinical trials investigating RP-1 (another HSV-1-based platform) in combination with nivolumab are showing promising results. The future of OV therapy lies in enhancing the tumor tropism and building the construct of the platform with checkpoint ligands. RP-2 and -3 are an effort in that direction, and trials are underway to investigate their efficacy in various cancers, including the rarer subsets of melanoma, like uveal melanoma

## Figures and Tables

**Figure 1 vaccines-13-00727-f001:**
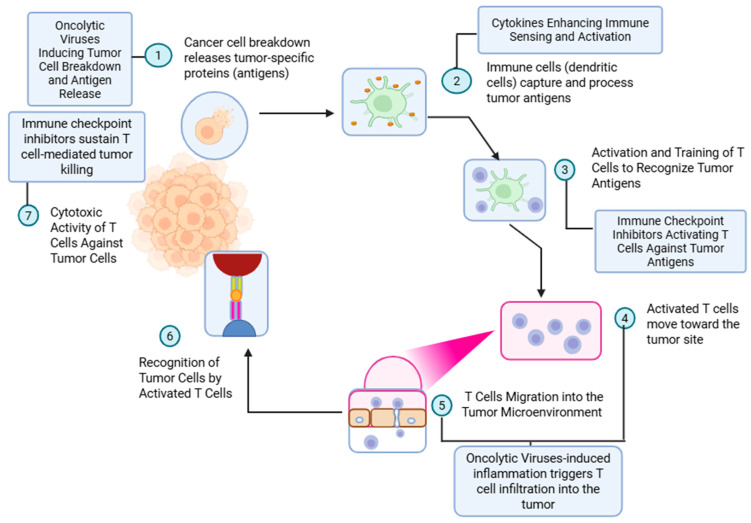
Mechanism of oncolytic virus-induced antitumor immunity and augmentation by immune checkpoint inhibitors. This schematic outlines the sequential immune events initiated by oncolytic virus therapy and potentiated by immune checkpoint blockade. (1) Oncolytic viruses infect and lyse tumor cells, resulting in the release of tumor-specific antigens. (2) These antigens are captured and processed by antigen-presenting cells (APCs), such as dendritic cells, in the presence of cytokines that enhance immune sensing and activation. (3) Processed antigens are presented to naïve T cells, leading to their activation and priming against tumor-associated antigens. This process is further enhanced by immune checkpoint inhibitors that prevent T-cell anergy and exhaustion. (4) Activated T cells migrate toward the tumor site. (5) Virus-induced inflammation promotes T-cell infiltration into the tumor microenvironment. (6) Infiltrating T cells recognize tumor cells via antigen–MHC complexes. (7) Ultimately, cytotoxic T cells mediate targeted tumor cell killing. Immune checkpoint inhibitors sustain this cytotoxic response, enhancing the efficacy of oncolytic virotherapy in tumor eradication.

**Table 1 vaccines-13-00727-t001:** List and status of ongoing trials in the United States between 2015 and May 2025 investigating novel oncolytic platforms in patients diagnosed with cutaneous and non-cutaneous melanoma.

Trial No.	Virus	Characteristics	Participants	Treatment Protocol	Best Response	Current Status
**Phase I/Ib trials**						
** *Double-stranded DNA viruses* **						
NCT03767348 [90]	HSV-1 (RP-1)	Expressing human GM-CSF and a fusogenic protein (GALV-GP-R−)	Multiple solid tumors	Monotherapy and in combination with nivolumab	156 patients ORR: 31.4% CR: 12.2%	Phase I/II study Led to phase III trial (ongoing IGNYTE-3 study)
NCT04336241 [92]	HSV-1 (RP-2)	Expresses GM-CSF, fusogenic protein GALV-GP-R-, and anti-CTLA-4 like molecule	Multiple solid tumors (including ICI-refractory uveal melanoma)	Monotherapy (IT) and in combination with nivolumab	Prelim results (uveal melanoma) 17 patients enrolled: Monotherapy: 3 patients Combination: 14 patients Combination arm: ORR: 28.6% (all PR) DCR: 57.1% Median DoR: 5.1 months	Recruiting
NCT04735978 [96]	HSV-1 (RP-3)	Expresses anti-CTLA-4 antibody, CD40 ligand, and h4-1BBL	Advanced or metastatic non-neurological solid tumors	Monotherapy (IT) or in combination with IV nivolumab	No results	Recruiting
NCT00429312 [98]	Vaccinia virus (JX-594)	Thymidine kinase-deleted vaccinia virus plus GM-CSF (*pexastimogene devacirepvec*)	Unresectable melanoma or metastatic melanoma	Monotherapy	10 patients enrolled PR: 2 patients, CR: 1 patient Injected tumor: 5/7 patients Non-injected tumors: 4/7 patients Median survival: 7.1 months	Not in development Tested in pre-ICI era
NCT06171178	Vaccinia virus (ASP1012)	Delivered IV expressing Leptin-IL2 as a payload	Multiple solid tumors including melanoma	Part 1: monotherapy then dose expansion in multiple tumor types	No results posted	Recruiting
NCT06444815	Vaccinia virus (VET-3-TGI)	Expresses CXCR3, IL-12, and a TGFB1-antagonizing mini-monomer	Multiple solid tumors	Groups A and C: monotherapy, IT and IV Groups B and D: IT or IV with pembrolizumab	No results posted	Recruiting
NCT05859074	Vaccinia virus, Ankara strain (MVA) (MQ710)	Induces cGAS/STING pathway and induces Type 1 IFNs	Multiple solid tumors	Monotherapy and in combination with pembrolizumab	No results posted	Recruiting
NCT05222932	Adenovirus (TILT-123)	Tumor Necrosis Factor Alpha- and IL-2-coding oncolytic adenovirus	HNSCC and melanoma	TILT-123 + Avelumab	No results posted	Recruiting
NCT05076760	Adenovirus (MEM-288)	Co-expresses MEM40 (CD40 ligand) and IFNβ	Multiple solid tumors including melanoma	Part 1: MEM-288 monotherapy Part 2: MEM-288 with nivolumab	No results posted	Recruiting
** *RNA viruses* **						
** *Positive-sense RNA viruses* **						
NCT03408587 [99]	Coxsackievirus A21 (CVA21)	Targets ICAM-1-expressing cancer cells (*Gebasaxturev*)	Uveal melanoma metastatic to liver	CVA21 + ipilimumab	11 patients enrolled SD: 3 patients. All patients had PD by week 26	Not in development (no meaningful responses)
NCT02307149 [100]	CVA21 (also known as V937)	Targets ICAM-1-expressing cancer cells	Unresectable melanoma or metastatic melanoma	CVA21 (IT) + ipilimumab	50 patients enrolled. ORR: 30% (47% in treatment naïve, 21% in anti-PD1 resistant) Median PFS: 6.2 months Median OS: 45.1 months	Phase Ib trial (MITCI study)
NCT02565992 [101]	CVA 21	--	Unresectable melanoma or metastatic melanoma	CVA21 (IT) + pembrolizumab	36 patients enrolled ORR: 47%, CR: 22%	Phase Ib study (CAPRA trial)
** *Negative-sense RNA viruses* **						
NCT03865212 [102]	Vesicular stomatitis virus	Includes genes for human IFN-β and TYPR1 (expressed in melanocytes and melanoma cells)	Metastatic uveal melanoma	Monotherapy—multiple dose levels	12 patients SD: 4 patients. PD: 8 patients Two patients treated with ICIs later on had durable responses	Not in development (no meaningful responses)
**Phase II/III trials**						
** *Double-stranded DNA viruses* **						
NCT03190824 (Phase IIA)	Adenovirus (OBP-301)	Replicates selectively in cancer cells by introducing hTERT promotor	Unresectable or metastatic melanoma	Monotherapy	--	Status unknown. Primarily developed in liver and GI cancers
NCT02272855 [103]	HSV-1 (HF-10, now called TBI-1401)	Mutated virus without any foreign genes	Ipilimumab-naïve unresectable melanoma	HF-10 + ipilimumab	46 patients enrolled ORR at 24 weeks: 41% SD: 68% Median PFS: 19 months Median OS: 21.8 months	Current development unknown in melanoma
NCT06581406	HSV-1 (RP-2)	Expresses GM-CSF, fusogenic protein GALV-GP-R-, and anti-CTLA-4-like molecule	ICI-naïve metastatic uveal melanoma	Arm 1: IT RP-2 + Nivolumab Arm 2: ipilimumab 3 mg/kg and nivolumab 1 mg/kg	Recruiting	Ongoing phase II/III open-label trial
NCT06264180	HSV-1 (RP-1)	Expressing human GM-CSF and a fusogenic protein (GALV-GP-R−)	Ipilimumab- and nivolumab-refractory and BRAF-MEK inhibitor-refractory cutaneous melanoma	Arm 1: nivolumab + RP-1 Arm 2: investigators’ choice (except TIL)	Recruiting	Randomized phase III trial
** *RNA viruses* **						
** *Positive-sense RNA viruses* **						
NCT01227551 [104]	CVA 21 (also known as V937)	--	Unresectable or metastatic melanoma	IT monotherapy	57 patients enrolled: all endpoints at 6 months PFS rate: 38.6% Durable ORR: 21.1% Best ORR: 28.1% 12: month PFS: 32.9% 12: month OS: 75.4%	Led to combination phase II study in combination with pembrolizumab
NCT04152863 (Phase II)	CVA 21 (also known as V937)	--	Unresectable or metastatic melanoma	CVA 21 either IV or IT with pembrolizumab IV	28 patients in each arm IV V937 + IV Pembro: 46.4% IT V937 + IV Pembro: 39.3% IV Pembro alone: 34.5%	Phase II trial terminated early due to business reasons (no publications)

Abbreviations: CR—complete response; CTLA-4—cytotoxic T lymphocyte-associated antigen 4; CVA21—Coxsackievirus strain A21; CXCR3—C-X-C motif chemokine receptor 3; DCR—disease control rate (CR + PR + SD); DoR—duration of response; GALV-GP-R—gibbon ape leukemia virus envelope glycoprotein with R-sequence deleted; GM-CSF—granulocyte monocyte colony stimulating factor; h4-1BBL—human 4-1BB ligand or CD137 ligand; HNSCC—head and neck squamous cell cancer; HSV-1—Herpes Simplex Virus 1; hTERT—human telomerase reverse transcriptase; ICI—immune checkpoint inhibitor; IFN-β—interferon beta; IL-12—interleukin-12; IL-2—interleukin-2; IT—intratumoral injection; IV—intravenous injection; ORR—overall response rate; PD—progressive disease; PR—partial response; SD—stable disease; TGFB-1—Transforming Growth Factor Beta 1; TIL—tumor-infiltrating lymphocyte therapy; TYPR-1—tyrosinase-related protein-1; V937—name of the strain of Coxsackievirus A21.

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
