# Peer review of "Advances in Oncolytic Viral Therapy in Melanoma: A Comprehensive Review"

_vaccines, 2025, doi:10.3390/vaccines13070727_

Round 1

Reviewer 1 Report

Comments and Suggestions for Authors

Garg et al intend to provide a review of oncolytic viral therapy in melanoma.

However, a significant part of the review (pp. 1-5, pp.11-14) is dedicated to historical aspects and mechanisms of action of oncolytic viruses in general. This part lacks novelty and is too superficial to have any value for the reader. It should be shortened and limited to studies related to the use of oncolytic viruses in melanoma.

Authors do not compare or analyze results of OV clinical trials in melanoma. An attempt to identify those approaches that improve efficacy of OV (in addition to combination with checkpoint inhibitors) would be beneficial. The mere listing of OV clinical trials in melanoma is not of interest. The discussion is absent; the conclusion is shallow and contains incorrect claims (e.g. promising results ща RP-3 in clinical trials).

The manuscript cannot be published in its present form due to abundance of inconsistencies in the text. In many occasions there are missing words or other deficiencies.

Examples (there are to many to list them all):

“A response was noted in all 35 injected lesions (11 CR and 24 PR) and 21 of 23 ??? lesions (12 PR and 9 CR)”

“with no remaining injectable ???”

“4.3. T-VEC in Ipilimumab”

The title of the Figure 1 does not correspond to the content of this figure.

The statement “Direct intratumoral injection of OV is better at generating anti-tumoral immune response compared to systemic immune checkpoint inhibitors (ICI)” is not supported by any example or citation.

Authors should comment or cite relevant sources to clarify what they meant by these statements, which otherwise seem to be erroneous:  

«abundance of aberrant nucleotides in tumor cells due to the defective oncogenic pathways provides a rich pool of genomic material for OVs to use and replicate»

“soluble tumor antigens are present, which are picked up by the T-cells to generate an antitumor response”

“oncolytic drive T-cell checkpoint molecules”

The statement that inflammatory response is the main result of the blockade of tumor vasculature should also be rephrased.

Author Response

Thank you for taking the time to review this manuscript and for your constructive criticism. We have prepared a response to your critiques. We have revised the manuscript to the best of our ability, incorporating some changes, and we hope that our responses will be satisfactory. We would also like to point out that this manuscript is not a systematic review or a meta-analysis, and we have not made such a claim anywhere. Hence, some of the critiques suggested by the reviewers cannot be incorporated into the manuscript. Thank you again for your time and critique.

Comment 1: Garg et al intend to provide a review of oncolytic viral therapy in melanoma. However, a significant part of the review (pp. 1-5, pp.11-14) is dedicated to historical aspects and mechanisms of action of oncolytic viruses in general. This part lacks novelty and is too superficial to have any value for the reader. It should be shortened and limited to studies related to the use of oncolytic viruses in melanoma.

Reply: Thank you for your comments. We humbly disagree with the reviewer’s assessment. The history of oncolytic viruses is crucial to the reader, as it traces the journey and the challenges encountered in the development of these drugs. We understand that we have spent a considerable amount of time on the history section, and it is still not a comprehensive review. However, we believe it is essential to present it in its current form, allowing the readers to appreciate the work that has gone into developing the OV over the last 100 years. If someone is not interested in reading it, then they can skip the next relevant section of their choice. We apologize for disagreeing with your comments. We will keep this section as it is. 

Comment 2: Authors do not compare or analyze results of OV clinical trials in melanoma. An attempt to identify those approaches that improve efficacy of OV (in addition to combination with checkpoint inhibitors) would be beneficial. The mere listing of OV clinical trials in melanoma is not of interest. The discussion is absent; the conclusion is shallow and contains incorrect claims (e.g. promising results ща RP-3 in clinical trials).

Reply: Thank you for that comment. We will take this opportunity to clarify that this manuscript is not a systematic review or meta-analysis, in which we would draw statistical comparisons between different trials. This is a narrative review. We have neither claimed nor attempted to do any statistical analysis in this manuscript. As a reviewer and author of several meta-analyses, I can confidently say that the heterogeneity among the trials is so high that a meaningful attempt at meta-analysis or systematic review will only lead to statistical jargon. Any such effort will only be a statistical exercise without any scientific basis.

Regarding the comment that listing the trials is not useful, we are not aware of any manuscript published within the last year that presents a comprehensive overview of all trials of OV-based products in melanoma, along with their results. Since this is a narrative review, we do not believe that any significant statistical conclusions can be drawn. The goal of the manuscript is to provide a comprehensive overall review that has been presented. Still, we have worked a little bit more on the conclusion part and rewritten it. If there is any particular aspect that the reviewer is looking for, then we are happy to consider that part.

Comment 3: The manuscript cannot be published in its present form due to the abundance of inconsistencies in the text. In many occasions there are missing words or other deficiencies.

Examples (there are to many to list them all):

“A response was noted in all 35 injected lesions (11 CR and 24 PR) and 21 of 23 ??? lesions (12 PR and 9 CR)”

“with no remaining injectable ???”

“4.3. T-VEC in Ipilimumab”

Reply: We have reviewed the literature again. Some of these were unfortunate errors that came during the grammar correction process. Many of these sections have been rewritten to reflect the correct interpretation of the data. We will be happy to address any other concerns or critiques.

Comment: The title of the Figure 1 does not correspond to the content of this figure.

Reply: The figure has been changed, and the title and legend have been added with appropriate explanations.  

The statement “Direct intratumoral injection of OV is better at generating anti-tumoral immune response compared to systemic immune checkpoint inhibitors (ICI)” is not supported by any example or citation.

Reply: reference 6 is the basis of this statement. This is the paper published in the nature group and is the basis of this statement.

Comment: Authors should comment or cite relevant sources to clarify what they meant by these statements, which otherwise seem to be erroneous:  

«abundance of aberrant nucleotides in tumor cells due to the defective oncogenic pathways provides a rich pool of genomic material for OVs to use and replicate»

Reply: rephrased and citation added.

“soluble tumor antigens are present, which are picked up by the T-cells to generate an antitumor response”

Reply: rephrased and citation added

“oncolytic drive T-cell checkpoint molecules”

Reply: rephrased and citation added.

The statement that the inflammatory response is the main result of the blockade of tumor vasculature should also be rephrased.

Reply: rephrased and citation added.

Reviewer 2 Report

Comments and Suggestions for Authors

Garg et al. presented in their review the current state of therapy for malignant melanoma using oncolytic viruses (OVs). Initially, they provided a historical overview of the use of viruses in cancer treatment and explained the mechanisms by which OVs exert their effects. They then discussed clinical studies in which OVs have been used for treatment of malignant melanoma, either as monotherapy or in combination with other therapies. Particular attention in these clinical studies was given to the OV T-VEC and the newer OVs (RP-1, RP-2, and RP-3). Finally, they categorized the different groups of OVs, briefly described their functions, and outlined their current applicability in melanoma treatment.

The study is very interesting, well-structured, and easy to read. It provides a comprehensive overview of the current state of knowledge regarding the use of OVs in malignant melanoma.

Minor points

  1. The conclusion is rather general. It would be advisable to support the efficacy of OVs (e.g., T-VEC) more clearly — for example, by stating the percentage of patients who benefit from OV monotherapy or combination therapies. Additionally, an outlook on potential future developments or strategies to further improve therapeutic success could enhance the conclusion.

  2. There are numerous abbreviations throughout the manuscript. These should be introduced upon their first mention (e.g., oncolytic viruses). The abbreviation should also be included in the abstract. Some abbreviations were introduced more than once (e.g., DC, on page 2 and page 5), and others were not explained at all (e.g., PD-L1). The authors should thoroughly review the manuscript in this regard.

  3. On page 6 (last paragraph), a reference [1] is cited. This appears to be incorrect.

Author Response

Thank you for taking the time to review this manuscript and for your constructive criticism. We have prepared a response to your critiques. We have revised the manuscript to the best of our ability, incorporating some changes, and we hope that our responses will be satisfactory. 

Comments: The conclusion is rather general. It would be advisable to support the efficacy of OVs (e.g., T-VEC) more clearly — for example, by stating the percentage of patients who benefit from OV monotherapy or combination therapies. Additionally, an outlook on potential future developments or strategies to further improve therapeutic success could enhance the conclusion.

Reply: Thank you for your comments. We are limited by the number of words in the conclusion paragraph; hence, we chose not to expand on the aspects we presented throughout the manuscript. Moreover, we cannot accurately determine the number or percentage of people who will benefit from OV therapy due to the negative randomized controlled trials in this area. However, we have included clinical scenarios where patients may benefit. We have also made adjustments to the conclusion to enhance the outlook, as per your suggestion. 

Comment 2: There are numerous abbreviations throughout the manuscript. These should be introduced upon their first mention (e.g., oncolytic viruses). The abbreviation should also be included in the abstract. Some abbreviations were introduced more than once (e.g., DC, on page 2 and page 5), and others were not explained at all (e.g., PD-L1). The authors should thoroughly review the manuscript in this regard.

Reply: We have reviewed all abbreviations and corrected the order as suggested. 

Comment: On page 6 (last paragraph), a reference [1] is cited. This appears to be incorrect.

Reply: Thank you for catching that. This is a proofreading error and has been rectified. 

Reviewer 3 Report

Comments and Suggestions for Authors

Very well written review that will be a great addition to the field of research on oncolytic viruses for the treatment of melanoma.

Minor comments:

1) There is inconsistency in writing T-VEC or TVEC throughout the manuscript, please review, the same is noted for ICP47 or ICP-47

2) Figure 1 could be improved by being more specific to oncolytic viruses, for example,  replace "Therapies" in box 1 by Oncolytic virues, replace "agents" in box 2 by cytokines, and "drugs" in box 3 by immune checkpoint inhibitors, etc.

Author Response

Reply to Reviewer 3

Thank you for taking the time to review this manuscript and for your constructive criticism. We have prepared a response to your critiques. We have revised the manuscript to the best of our ability, incorporating some changes, and we hope that our responses will be satisfactory.

Comment 1: There is inconsistency in writing T-VEC or TVEC throughout the manuscript, please review, the same is noted for ICP47 or ICP-47

Reply- We have carefully combed through the manuscript, and the abbreviations have been corrected

Comment 2: Figure 1 could be improved by being more specific to oncolytic viruses, for example,  replace "Therapies" in box 1 by Oncolytic virues, replace "agents" in box 2 by cytokines, and "drugs" in box 3 by immune checkpoint inhibitors, etc.

Reply: Thank you for the suggestion. We have completely changed the outlook of Figure 1 and the legend to make it suitable for the manuscript. Hopefully, this will serve the purpose better.  

Reviewer 4 Report

Comments and Suggestions for Authors

This manuscript provides a well-organized and comprehensive review of oncolytic viral therapies in melanoma, spanning both historical milestones and recent clinical developments. The authors have clearly invested significant efforts in compiling a broad and up-to-date summary of the field. This review is thorough, with strong emphasis on key agents such as T-VEC, RP-1, and newer constructs like RP-2 and RP-3, making this review particularly valuable for clinicians and researchers tracking advances in OV-based immunotherapy in melonoma.

The overall structure is logical and effective, helping readers easily navigate complex biological mechanisms and clinical data. That being said, I have a few suggestions that could help clarify key points and enhance the manuscript’s impact:

  1. Syncytia Formation and Selectivity
    While the authors note that GALV-GP-R− enhances tumor cell fusion and lysis, the mechanism of the GALV and syncytia formation is not well explained. Besides, it would be helpful to clarify whether the fusogenic activity is strictly confined to tumor cells, or if it could affect adjacent normal cells.
  2. Checkpoint Inhibitor Distinctions
    The manuscript describes several combination trials involving checkpoint inhibitors such as nivolumab, pembrolizumab, and ipilimumab. However, it may be helpful to briefly discuss their mechanisms of action (e.g., PD-1 vs. CTLA-4 blockade) and reflect on why certain combinations (e.g., T-VEC + pembrolizumab) did not meet clinical endpoints, whereas others (e.g., RP-1 + nivolumab) appear more promising.
  3. Figure 1 – Title and Content
    The current title of Figure 1 ("Mechanism of Action of Oncolytic Viruses") is somewhat misleading, as the illustration more closely depicts a general immune activation cascade rather than the specific mechanisms of OV therapy (e.g., viral infection, replication, lysis, or syncytia formation). I recommend rephrasing the title to something more accurate. Additionally, the figure would benefit from notations or a legend explaining the components and sequence of events. Including OV-specific features (for example, viral entry or oncolysis) would help strengthen the figure’s relevance to the subject matter.
  4. Table 1 –  Formatting
    In Table 1, the phrase "from 2015 to now" may become unclear or outdated for readers in future years. Consider revising to a more neutral and enduring phrasing, such as “from 2015 to the present (as of May 2025)” 
    Additionally, Table 1 is information-rich but visually dense. Simple enhancements like color-coding, or grouping by virus platform could improve readability and highlight key findings more effectively. Clinical trial 1-2 and 3 maybe need more space to be separated. Also, could the authors clarify the ordering of clinical trials in Table 1?
  5. Minor Consistency Issue
    The manuscript uses both "T-VEC" and "TVEC" inconsistently. Please standardize the abbreviation. “T-VEC” appears to be the most widely accepted and should be used consistently throughout.

Conclusion:
This is a useful and timely review with minor refinements needed to improve clarity and consistency. It will be a valuable resource for the field of melanoma oncolytic immunotherapy.

Author Response

Reply to Reviewer 4

Thank you for taking the time to review this manuscript and for your constructive criticism. We have prepared a response to your critiques. We have revised the manuscript to the best of our ability, incorporating some changes, and we hope that our responses will be satisfactory.

Comment: Syncytia Formation and Selectivity: While the authors note that GALV-GP-R− enhances tumor cell fusion and lysis, the mechanism of the GALV and syncytia formation is not well explained. Besides, it would be helpful to clarify whether the fusogenic activity is strictly confined to tumor cells or if it could affect adjacent normal cells.

Reply: Thank you for the suggestion. We have added an explanation regarding the mechanism of action of fusogenic proteins in the manuscript (Page 17). We did not find any reference confirming or denying the role of fusogenic proteins in increasing the infectivity of normal cells. We believe that there is no direct connection between the expression of fusogenic proteins and the infective potential of the OV towards normal cells; however, we are happy to review any relevant literature if you are aware of any. 

Comment 2: Checkpoint Inhibitor Distinctions: The manuscript describes several combination trials involving checkpoint inhibitors such as nivolumab, pembrolizumab, and ipilimumab. However, it may be helpful to briefly discuss their mechanisms of action (e.g., PD-1 vs. CTLA-4 blockade) and reflect on why certain combinations (e.g., T-VEC + pembrolizumab) did not meet clinical endpoints, whereas others (e.g., RP-1 + nivolumab) appear more promising.

Reply: We value your suggestion and agree with your perspective. However, we think that such a discussion is beyond the scope of this manuscript. We have updated Figure 1, detailing the mechanism of action of oncolytic viruses and how they can contribute to increasing the immunogenicity of tumors. Our goal is to provide a comprehensive overview of the journey of oncolytic viral therapy. The mechanism of action of various aspects is a very detailed discussion, which, in our humble opinion, is beyond the purview of this manuscript.  

Comment 3: Figure 1 – Title and Content: The current title of Figure 1 ("Mechanism of Action of Oncolytic Viruses") is somewhat misleading, as the illustration more closely depicts a general immune activation cascade rather than the specific mechanisms of OV therapy (e.g., viral infection, replication, lysis, or syncytia formation). I recommend rephrasing the title to something more accurate. Additionally, the figure would benefit from notations or a legend explaining the components and sequence of events. Including OV-specific features (for example, viral entry or oncolysis) would help strengthen the figure’s relevance to the subject matter.

Reply: We agree with this suggestion. We have updated Figure 1 and added a legend.

Comment 4: Table 1 –  Formatting: In Table 1, the phrase "from 2015 to now" may become unclear or outdated for readers in future years. Consider revising to a more neutral and enduring phrasing, such as “from 2015 to the present (as of May 2025)” 
Additionally, Table 1 is information-rich but visually dense. Simple enhancements like color-coding or grouping by virus platform could improve readability and highlight key findings more effectively. Clinical trials 1-2 and 3 may need more space to be separated. Also, could the authors clarify the ordering of clinical trials in Table 1?

Reply: We agree with your suggestion and have adjusted the timeline accordingly in the title. Unfortunately, we do not think color coding will help as the tables are published in black and white, to the best of our understanding. There is no particular order for listing clinical trials. We have separated and listed trials as phases I, II, and III. There is no specific order to the trials listed, as the information was gathered according to the product's evaluation phase. 

Comment 5: Minor Consistency Issue: The manuscript uses both "T-VEC" and "TVEC" inconsistently. Please standardize the abbreviation. “T-VEC” appears to be the most widely accepted and should be used consistently throughout.

Reply - We have reviewed all abbreviations and corrected these errors. 

Round 2

Reviewer 1 Report

Comments and Suggestions for Authors

I thank authors for correcting inconsistencies. The quality of the manuscript has been significantly improved. However, I still think that this paper would have benefited from an attempt to analyze those factors that may have an impact on the efficacy of OV therapy in melanoma and to identify directions for further research in addition to combination with checkpoint inhibitors.

Author Response

Comment: I thank authors for correcting inconsistencies. The quality of the manuscript has been significantly improved. However, I still think that this paper would have benefited from an attempt to analyze those factors that may have an impact on the efficacy of OV therapy in melanoma and to identify directions for further research in addition to combination with checkpoint inhibitors

Reply: We again thank the reviewer for his valuable time and suggestions. We deeply respect your opinion and experience. 

We would like to reiterate that our intention is not to analyze the data that has been presented. That process is different, and the literature review for that process would entail a different kind of effort (meaning literature search including small prospective studies, validation of studies, inclusion and exclusion criteria, accounting for heterogeneity, etc.). That is not within the scope of this manuscript, and we do not want to alter the manuscript's focus at this point. Hence, we would humbly decline this request. 

Regarding the comment on future directions of research, we believe that this is a topic in itself, which warrants discussion at the level of translational science. This is also beyond the purview of this manuscript. Our goal is to present clinical data and review the viruses that can be used as a vehicle. We also discuss ongoing studies involving these viruses. In line with this goal, we discussed RP-2 and RP-3, referencing the publicly available data at this point. Hence, we would humbly decline to delve deeper into the developmental aspects of the oncolytic viral platform.